# A New Monitoring Technology for Bearing Fault Detection in High-Speed Trains

**DOI:** 10.3390/s23146392

**Published:** 2023-07-14

**Authors:** Sitong Sun, Sheng Zhang, Wilson Wang

**Affiliations:** 1School of Automation and Electronic Engineering, Qingdao University of Science and Technology, Qingdao 266000, China; 2021040025@mails.qust.edu.cn; 2Department of Mechanical and Mechatronics Engineering, Lakehead University, Thunder Bay, ON P7B 5E1, Canada; wilson.wang@lakeheadu.ca

**Keywords:** vibration signal collection, bearing condition monitoring, variational mode decomposition, multiple correlation analysis

## Abstract

In this work, a new monitoring system is developed for bearing fault detection in high-speed trains. Firstly, a data acquisition system is developed to collect vibration and other related signals wirelessly. Secondly, a new multiple correlation analysis (MCA) technique is proposed for bearing fault detection. The MCA technique consists of the three processing steps: (1) the collected vibration signal is decomposed by variational modal decomposition (VMD) to formulate the representative intrinsic mode functions (IMFs); (2) the MCA is used to process and identify the characteristic features for signal analysis; (3) bearing fault is diagnosed by examining bearing characteristic frequency information on the envelope power spectrum. The effectiveness of the proposed MCA fault detection technique is verified by experimental tests corresponding to different bearing conditions.

## 1. Introduction

High-speed trains have become one of the most commonly used transportation means for people traveling, especially in China. Thus, safety and reliability are the most important issues in high-speed train operations [1,2]. Reliable train condition monitoring, however, still remains a challenging task in research and development [3,4]. Among the mechanical transmission systems in a high-speed train, rolling element bearings of the wheel shafts can have the highest failure rate. Bearing defects may lead to machine performance degradation or early equipment failure. Correspondingly, reliable bearing fault detection techniques are very helpful to recognize a bearing defect at its earliest stage so as to improve high-speed train performance and prevent malfunction or even unexpected accidents [5].

Bearing fault detection can be undertaken by analysis of several types of signals such as vibration, electric current/voltage, and lubricant [6]. Vibration-based analysis can be the most commonly used approach due to its ease of measurement and higher signal-to-noise ratio [7], which will also be used in this work. Although there are data acquisition (DAQ) systems commercially available for vibration measurement [8], they may not be suitable for direct high-speed train applications; correspondingly, the first objective of this work is to develop a specific DAQ system for signal collection wirelessly.

Reliable bearing fault detection could be a more challenging task in comparison with other rotating units like gears and shafts, as a rolling element bearing is a system, instead of a component like a gear or shaft; a bearing consists of several components such as the outer ring, inner ring, and rolling elements. Furthermore, fault-related features are usually weak and modulated by other strong vitiations and noise, as well as machine operating conditions [9,10].

Bearing fault detection can be undertaken using model-based approaches such as convex system analysis [11], and signal processing-based techniques. Each hasits own merits and limitations. Signal processing approach is used in this work. Bearing fault detection based on signal processing can be undertaken in the time domain, frequency domain, or time–frequency domain [12]. Research results have found that direct time and statistical analysis may not be able to generate consistent indicators for bearing fault detection, especially under variable operating conditions [13]. In the frequency domain, the spectral analysis may not be appropriate for bearing fault diagnosis if the signatures involve time varying components, such as the case when fault occurs on the rotating bearing components [14,15]. The classical time–frequency analysis techniques include the short-time Fourier transform, continuous/discrete wavelet transform, Hilbert–Huang transform, etc. [16]. Variational mode decomposition (VMD) is a new method for decomposing a signal into multiple signatures with different center frequencies to detect bearing faults. It can suppress modal aliasing and endpoint effects in empirical mode decomposition (EMD) [17]. In processing, the envelope entropy can be used as the fitness function for VMD parameter optimization. However, the general VMD lacks adaptive capability, where the fault features are extracted manually [18]. Jin et al. [19] proposed a multi-objective VMD parameter optimization for bearing fault detection; however, in feature extraction, the extracted intrinsic mode functions(IMFs) may not be sufficient to identify bearing faults with different system conditions. Wei et al. [20] suggested a VMD method, in which VMD parameters are optimized by envelope entropy analysis; Wu et al. [21] used center frequency statistical analysis to determine the number of intrinsic functions in VMD; however, these methods are computationally complex and may not be suitable for projects in real-time processing. Zhang et al. [22] improved VMD by using resonance demodulation for fault detection of locomotive bearings; however, the method of selecting IMFs is based on the maximum principle, which may not be always representative. He et al. [23] proposed an adaptive VMD method for feature extraction and multiple faults detection in a rotating machinery; however, this method has alimitation of extracting weak fault features. On the other hand, a few methods were suggested in the literature for bearing fault detection based on the analysis of adaptive spectral kurtosis [24], envelope spectrum L-kurtosis [25,26], and entropy [27,28,29,30]. However, the recognized characteristic features using these methods may not be sensitive to system and test conditions and may not be effective for bearing fault detection.

To tackle the aforementioned problems, the second objective of this work is to propose a multiple correlation analysis (MCA) technique for bearing fault detection. The contributions in this work include the following aspects: (1) A new DAQ system is developed to correct vibration and other related signals wirelessly in a high-speed train. (2) In the proposed MCA technique, the vibration signal is modulated by using a novel VMD approach including mutual and envelope correlation analysis to recognize representative features. (3) The spectrum–kurtosis analysis is applied for bearing fault detection. The effectiveness of the proposed technology is verified by experimental tests.

The rest of this paper is organized as follows. The developed DAQ system is presented in Section 2. The proposed MCA technique is discussed in Section 3. The effectiveness of the proposed technique is verified experimentally in Section 4. Some concluding remarks are summarized in Section 5.

## 2. Sensor Array Design and Impact Point Positioning

### 2.1. The Process of Signal Acquisition

A new DAQ system is specifically developed to collect signals in high-speed bullet train shaft systems. Figure 1 shows the block diagram of the DAQ system.

This DAQ system is used to collect signals from the train wheel shafts. The collected signals include vibration, instantaneous speed, and pressure by the use of appropriate sensors. The DAQ card is mainly used for signal conditioning. After the acceleration signal and velocity signal have been collected by the acceleration sensor and velocity sensor, the signals are sent to the signal conditioning board for signal conditioning.

#### Sensor Selection

The selected acceleration sensors are used to measure shaft vibrations. They are FA101-A3-±2g-LC13 piezoresistive accelerators (made by Tyco Electronics Connectivity, Qingdao, China), which have frequency bandwidths of 20 kHz, a measurement range of ±2g, and a sensitivity of 100 mv/g.

The instantaneous train speed is measured by using Doppler radar sensors (DRS6/1ac), which have frequency bandwidths of 19.2 kHz, a measurement range of 0~300 km/h, and a sensitivity of 200 mv/mm/s. The sensor output is fed to the transmitter via an RS232 protocol with a minimum transmission rate of 100 b/s.

The pressure sensors MPM4730 (made by Microsensor, Baoji, China) are used to measure changes in train pressure during braking. They are mounted on the train suspension and have a measurement range of 0~0.7 MPa, a comprehensive accuracy of ±0.15% FS, and a long-term stability of ±0.2% FS/year.

Figure 2 shows the mainframe structure. It consists of the signal conditioning board, the DAQ card, the wireless transmitter module, and the CPU mastering.The DAQ card uses PCI-6221, an 8-channel bidirectional channel, with a minimum voltage range sensitivity of 5.2 uF and a sampling rate of 250 kS/s. The wireless transmitter module and the signal conditioning board are described in the following subsections.

### 2.2. Wireless Transmitter Modules

After the signals are collected by using the related sensors, they are transmitted to the wireless transmitter via the RS232 interface. The wireless transmitter module is designed to operate within the ZigBee protocol with baud rate of 250 ksps and a receiving sensitivity of 100 dBm (1%). Figure 3 illustrates the schematic diagram of the developed DAQ module. The MAX3232 chip (manufactured by MAXIM) is for RS232 communication, and converts the logic signals of the microcontroller to the levels of RS232. MAX3232 can match 5 V systems to the RS232 interface, and interconnect 3 V systems to the RS232 interface (used in this system).The AMS1117 is a low dropout linear regulator with a voltage drop of 1.2 V at 1Aand has a very low self-consumption. It also features over-current protection, over-temperature protection, precision reference source, differential amplification, and time delay. The main controller of the embedded wireless DAQ module is the STM32 (based on the ARM Cortex-M3 core), which has properties such as powerful performance, fast response, low power consumption, and high compatibility.

The developed PCB board is shown in Figure 4. It is used for both the embedded wireless acquisition module and the embedded wireless collection module. The STM32 microcontroller operates at 3.3 V, which is obtained from the 12 V power supply via the voltage regulator chip AMS1117-3.3. Four pressure signals after the sampling are fed to the ADC inputs of the STM32 chip via a 160 Ω precision resistor.

#### Signal Conditioning Circuit

Because of the specific DAQ requirements of accuracy, linearity, and stability, an analog photoelectric isolation method is suggested for signal conditioning. The analog signal port of the DAQ card (PCI-6221, from Native Instruments, Qingdao, China) is processed and displayed by an upper computer Labview program. Figure 5 illustrates the schematic diagram of the signal conditioning and Figure 6 shows the developed PCB hardware.

The LM358 is a DC-coupled, low power, and high-gain dual operational amplifier, which has independent high-gain internal frequency compensation. It has a single power supply and can operate at supply voltages ranging 3.0~32 V.

Chip HCNR200 (by Agilent Enterprises, Beijing, China) is a high linearity analog photocoupler, which is used to isolate analog signals. It is composed of a high-performance LED and two photodiodes. Two photodiodes receive the signal output from the LEDs.

### 2.3. Implementation and Field Test

The developed DAQ system has been implemented in Lab view and primarily tested on existing trains. Figure 7 shows one of the tests in a high-speed train, and some of the hardware DAQ system are shown in Figure 8 and Figure 9. Figure 10 shows some examples of DAQ processes.

The following section will discuss the proposed shaft bearing fault detection technique by using the collected vibration signals only.

## 3. The Proposed MCA Technique for Bearing Fault Detection

### 3.1. Overview

The proposed MCA technique consists of the following processing procedures as illustrated in Figure 11.

(1)Decompose the collected vibration signal into the upper computer conditioning program; and then formulate a group of IMFs by using the VMD.(2)Compute the normalized correlation measure (NCM), the mutual information analysis (MIA), and the envelope correlation spectrum (ECS) for each IMF.(3)Calculate the cumulation values and average values of NCM, MIA, and ECS. If the cumulation value is less than the average, the corresponding IMF will be discarded. If the cumulation value is more than the average value, the IMF will be used to formulate an analytical signal.(4)Conduct spectral analysis of the analytical signal for bearing fault detection.

### 3.2. Variational Mode Decomposition

The VMD can suppress the mode aliasing of the EMD method by controlling the bandwidth of each decomposition iteratively [31]. It uses a non-recursive method to search a set of model components and their respective center frequencies. Each mode becomes smooth after demodulation to the baseband, and less sensitive to sampling and noise [32].

The constrained variational model involved in the VMD algorithm is:(1)$$\left\{\begin{array}{c} \underset{\left\{u_{k}\right\},\left\{\omega_{k}\right\}}{\min}\left\{\sum_{k}{\left\|\partial_{t}\left[\left(\delta \left(t\right)+j/\pi t\right)*u_{k}\left(t\right)\right]e^{-j\omega_{k}t}\right\|}_{2}^{2}\right\}\\ \mathrm{Constraints}:\;s.t.\sum_{k=1}^{K}u_{k}=f \end{array}\right.$$
where K is the number of modes to be decomposed; $\left\{u_{k}\right\}$ and $\left\{\omega_{k}\right\}$ are the *k*th mode component and center frequency after decomposition, respectively; δ(t) is the unit impulse function; j is the complex units; * represents a convolution operation; $\partial_{t}$ denotes partial derivative operation with respect to *t*; and f is the target signal.

To solve Equation (1), the Lagrange multiplier can be used to transform the constrained variational problem into an unconstrained variational problem, or:(2)$$L\left(\left\{u_{k}\right\},\left\{\omega_{k}\right\},\lambda \right)=\alpha {\sum_{k}\left\|\partial_{t}\left[\left(\delta \left(t\right)+j/\pi t\right)*u_{k}\left(t\right)\right]e^{-j\omega_{k}t}\right\|}_{2}^{2}+{\left\|f\left(t\right)-\sum_{k}u_{k}\left(t\right)\right\|}_{2}^{2}+\langle \lambda \left(t\right),f\left(t\right)-\sum_{k}u_{k}\left(t\right)\rangle$$
where α is the quadratic penalty factor, which is used to reduce the interference of Gaussian noise.

By using the alternating direction, the multiplier iterative algorithm, in combination with Parseval/Plancherel and Fourier isothermal transform, the modal components $u_{k}$, center frequencies $\omega_{k}$, and the saddle points of the augmented Lagrange function λ, can be iteratively optimized by:(3)$${\hat{u}}_{k}^{n+1}\left(\omega \right)\leftarrow \frac{\hat{f}\left(\omega \right)-\sum_{i\neq k}{\hat{u}}_{i}\left(\omega \right)+\hat{\lambda }\left(\omega \right)/2}{1+2\alpha {\left(\omega -\omega_{k}\right)}^{2}}$$
(4)$$\omega_{k}^{n+1}\leftarrow \frac{\int_{0}^{\infty }\omega {\left|{\hat{u}}_{k}^{n+1}\left(\omega \right)\right|}^{2}d\omega }{\int_{0}^{\infty }{\left|{\hat{u}}_{k}^{n+1}\left(\omega \right)\right|}^{2}d\omega }$$
(5)$${\hat{\lambda }}^{n+1}\left(\omega \right)\leftarrow {\hat{\lambda }}^{n}\left(\omega \right)+\gamma \left(\hat{f}\left(\omega \right)-\sum_{k}{\hat{u}}_{k}^{n+1}\left(\omega \right)\right)$$
where γ is the noise tolerance, which meets the fidelity requirement of signal decomposition; ${\hat{u}}_{k}^{n+1}\left(\omega \right)$, ${\hat{u}}_{i}\left(\omega \right)$, $\hat{f}\left(\omega \right)$, and $\hat{\lambda }\left(\omega \right)$ are the Fourier transform of $u_{k}^{n+1}\left(t\right)$, $u_{i}\left(t\right)$, f(t), and λ(t), respectively.

The main iterative processes of VMD are summarized as follows:(1)Initialize ${\hat{u}}_{k}^{1}$, $\omega_{k}^{1}$, $\lambda^{1}$, and the maximum number of iterations *N*, n←0;(2)Use Equations (3) and (4) to update ${\hat{u}}_{k}$ and $\omega_{k}$;(3)Use Equation (5) to update $\hat{\lambda }$;(4)The criterion of precision convergence is ε>0, where ε is a small positive number over (0, 0.01] (or up to 1%). ε = 0.001 is used in this work. If $\sum_{k}{\left\|{\hat{u}}_{k}^{n+1}-{\hat{u}}_{k}^{n}\right\|}_{2}^{2}/{\left\|{\hat{u}}_{k}^{n}\right\|}_{2}^{2}<\varepsilon$ or n=N the iteration is completed. Otherwise return to Step (2) and continue until the convergence conditions are satisfied.

### 3.3. MCA for IMF Selection

Each IMF satisfies the following two conditions [33]:(1)The number of local extrema (minima and maxima) and the number of zero crossings must either be equal or differ at most by one in the whole data set;(2)The mean value of the envelope defined by the local extrema is zero.

In this work, a novel IMF selection approach is proposed based on the NCM, MIA, and ECS analysis. The related processing is discussed as follows.

#### 3.3.1. Normalized Correlation Measure (NCM)

The NCM is an indicator for a linear similarity measure between two waveforms, which can be defined as:(6)$$C_{M}=\sum_{i=1}^{N}\left||R(y,x)|\right|/L_{N}$$
where $R(y,x)=E[(y\;-E(y))(x\;-E(x))]/(\sigma_{y}\sigma_{x})$ is the cross-correlation of the residual of the VMD output y and the original x; $L_{N}=2N-1$ and *N* are the length of the original signal; E[.] is the expect operator; $\sigma_{y}$ and $\sigma_{x}$ are the standard deviations of y and x, respectively.

#### 3.3.2. Mutual Information Analysis (MIA)

The MIA indicator aims to provide a nonlinear similarity measure to characterize representative features in the IMFs, which is defined as:(7)$$M_{I}(x,y)=2I(x,y)/(H(x)+H(y))$$
where $I\left(x,y\right)=\sum_{x}\sum_{y}p\left(x,y\right)\log\left(p\left(x,y\right)/\left(p\left(x\right)p\left(y\right)\right)\right)$ is a mutual information; *y* is the output of the VMD technique; *x* is the original signal. p(x,y) is the joint distribution; p(.) is the edge distribution; $H(x)=-\sum_{i}p(x_{i})\log(x_{i})$ is the information entropy.

#### 3.3.3. Envelope Correlation Spectrum (ECS)

The ECS indicator $I_{E}$ is applied to measure the correlation between the envelope spectra of the input signal x(t) and an IMF component, which is defined as:(8)$$I_{E}=R\left(F[V(x)],F[V(I_{n})]\right)$$
where R[.] represents the correlation operation; F[.] is the FT; and $I_{n}$ is the IMF component of the VMD decomposition. The envelope of the input signal x(t) is determined by:(9)$$V\left(x\left(t\right)\right)=\sqrt{\left(x{\left(t\right)}^{2}+{\left(H\left[x\left(t\right)\right]\right)}^{2}\right)}$$
where *H*(.) represents the Hilbert transform.

The cross-correlation between a signal p and a signal q is calculated by:(10)$$R\{p,q\}=E[(p-E(p))(q-E(q))]/\sigma_{p}\sigma_{q}$$
where $\sigma_{p}$ and $\sigma_{q}$ are the standard deviations of signals *p* and *q*, respectively.

MIA is considered as a measure of information of one random variable contained in another random variable. Normalization is applied so that the input signal for all samples has a zero mean compared to its mean squared deviation. The envelope spectrum is obtained by undertaking the Hilbert transform to the signal, taking all local extremes and the envelope of the resulting one-dimensional data, and then doing the spectral analysis of the envelope signature. This envelope analysis can reduce frequency interference and highlight the characteristic features in the signal, which can be used for bearing fault detection as discussed in Section 4.

### 3.4. Signal Reconstruction

The new IMF indicator $IMF_{n}$ is defined as:(11)$$IMF_{n}=a_{n}C_{M}(n)+b_{n}M_{I}(n)+c_{n}I_{E}(n)$$
where $a_{n}$, $b_{n}$, and $c_{n}$ are coefficients associated with specific applications. Each IMF, whose indicator is more than the average of all IMFs, is considered to contain representative information for advanced processing. If *N*IMFs are selected, the analytical signal X is formulated by:(12)$$X=\sum_{n=1}^{N}K_{u}(IMF_{n})\times IMF_{n}$$
where $K_{u}(.)$ is the kurtosis of the related IMF function.

## 4. Performance Evaluation Testing

### 4.1. Experimental Setup

The effectiveness of the proposed MCA technique will be examined experimentally corresponding to different bearing conditions. Figure 12 shows the experimental setup used in this work. The system is driven by a 3 hp induction motor, with aspeed range from 20 to 4200 rpm, controlled by a variable frequency drive (VFD022B21A). The variable load (torque) is applied by a magnetic brake through a bevel gearbox and a belt drive. An optical sensor is used to provide a one-pulse-per-revolution signal for shaft speed measurement. The bearing (MB ER-10K) on the left-hand side of the housing is for testing. The related bearing paraments are summarized in Table 1.

Four bearing health conditions are considered for the testing: healthy, bearing with an outer race fault, bearing with an inner race fault, and bearing with a rolling element defect. Table 2 summarizes the corresponding characteristic frequencies in terms of shaft speed *f_r_*.

To examine the effectiveness of the proposed MCA technique, a few related techniques will be for comparison:(1)The first technique for comparison is the Synchronous Influence Index (SII) method [23], which can assess the complex impulsive fault component, will be applied for comparison, as specified as SII;(2)The second method used for comparison is the frequency band entropy (FBE) analysis technique [17]. Its IMF section is based on an optimal algorithm that contains abundant fault information;(3)The third method for comparison is the general Hilbert–Huang (HH) transform technique. It uses the common IMF selection method or to choose the first two IMFs for analysis, which is designated as HH-C;(4)The fourth method used for comparison uses the general HH transform technique, but using kurtosis to choose two IMFs (with the highest kurtosis values) for analysis, designated as HH-K;(5)The fifth method applied for comparison is designated as NME, which uses the highest correlation values of the sum of the NCM, MIA, and ECS values, rather than the comprehensive assessment of these values by using the proposed MCA technique.

### 4.2. MCA Technique Implementation

Firstly, some examples are used to illustrate the IMF section approach in the proposed MCA technique. Figure 13a–d show IMFs obtained without using the proposed MCA technique, and Figure 13e–h are those using the proposed MCA selection method, corresponding to healthy bearings and bearings with outer race defect, inner race defect and rolling element defect, respectively.It is seen that the two IMFs with the highest magnitudes may not be the first two IMFs, instead, for example, IMF-4 and IMF-5 in Figure 13a. The proposed MCA technique can comprehensively consider the related effects and choose those with the most distinctive IMFs to formulate the analytical signal for advanced analysis.

Figure 14 shows the comparison between the reconstructed analytical signal and the original signal, corresponding to different bearing health conditions (i.e., healthy bearing, bearings with outer race defect, inner race defect, and rolling element defect), respectively. It can be seen that the reconstructed signals Figure 14a–d have similar characteristics of the original signal in Figure 14e–h, but with less noise or a higher signal-to-noise ratio.

### 4.3. Experimental Results Discussion

Some data sets corresponding to a shaft speed of 1797 rpm or *f_r_* = 30 Hz will be used for illustration.

#### 4.3.1. Healthy Bearing Condition Monitoring

Firstly, the bearings with health conditions are tested, with the characteristic frequency of *f_r_* = 30 Hz. Figure 15 shows the processing results for a healthy bearing using the related techniques. It can see that all of the related techniques can recognize the characteristic frequency components. However, the proposed MCA technique (Figure 15f) can provide the best diagnostic accuracy with the highest spectral magnitude corresponding to the characteristic frequency and its harmonics. Although the FBE technique in Figure 15b and the HH-K technique in Figure 15d can predict the bearing health condition in this case, the spectral maps contain many noise peaks due to IMF selection processes.

#### 4.3.2. Outer Race Fault Detection

Figure 16 shows the processing results using the related methods, with a characteristic frequency *f_r_* = 107 Hz. It can be seen from Figure 16 that all of the related techniques can recognize the defect on the outer ring. When a defect occurs on the fixed ring race of a bearing (the outer race in this case), its defect-induced resonance modes do not change over time, which is relatively easier to defect using the related signal processing techniques. However, the proposed MCA technique can not only extract a representative feature (the characteristic frequency) with the highest magnitude, but also provide clear harmonics, as demonstrated in Figure 16f, due to its efficient IMF section and processing procedures. Clear harmonics can also be used to confirm the fault detection results and improve diagnostic reliability. On the other hand, it is seen that although the BFE technique (Figure 16b) and NN-K method (Figure 16d) can predict the bearing fault, their magnitudes are much lower than using other related techniques, which can demonstrate the importance of using advanced IMF section approaches.

#### 4.3.3. Inner Race Fault Detection

Figure 17 shows the processing results for a bearing with an inner race defect with a characteristic frequency *f_r_* = 162 Hz in this case. The detection of a bearing fault in the rotating inner race is usually more challenging than the detection of a fault in the fixed outer race because the inner race rotates, and its resonance modes vary over time. In this case, it is clear from Figure 17f that the proposed MCA technique can provide the best fault diagnosis showing a dominant spectral component with a highest amplitude on the spectrum. Although the bearing fault characteristic frequency can be recognized by the SII (Figure 17a), the HH-C technique (Figure 17c), and the HH-K (Figure 17d) technique, these characteristic frequency components do not dominant the spectral maps, or they cannot provide reliable diagnostic results. The FEB and NME techniques can generate dominant characteristic frequency components due to their efficient processing to improve the signal-to-noise ratios; however, these spectral magnitudes are still lower than that processing by using the proposed MCA technique.

#### 4.3.4. Rolling Element (Ball) Defect Detection

The detection of a fault on the rolling element (i.e., ball in this case) is considered the most challenging task in bearing fault detection. This is because the rolling element rotates along different directions (as well as its slides), and its resonance modes change over time. Figure 18 shows the processing results for a bearing with a rolling element fault, with the characteristic frequency. In this case, none of these techniques can provide reliable diagnostic information about the bearing health condition; however, the proposed MCA technique (Figure 18f) still outperforms other related techniques by the use of the MCA approach to improve the signal-to-noise ratio and highlight characteristic frequency components, which have higher spectral magnitude.

## 5. Conclusions

Reliable fault detection in rolling element bearings still remains a challenging task in both research and applications, as the characteristic features of bearing faults are relatively weak and modulated by noise. The objective of this work is to develop a new monitoring technology for bearing fault detection in high-speed trains. Firstly, a new DAQ system is developed for this specific train shaft monitoring applications. Secondly, a new MCA technique is suggested for train shaft bearing fault detection based on analysis of vibration signals. The MCA takes a few processing steps: (1) the collected vibration signal is decomposed by the use of the VMD algorithm to formulate the representative IMFs; (2) a novel IMF selection approach is proposed based on the suggested NCM, MIA, and ESC analysis. The effectiveness of the proposed MCA fault detection technique has been verified by a series of experimental tests corresponding to different bearing health conditions. Test results have shown that the developed DAQ system can collect vibration and other related signals reliably. The proposed MCA technique can recognize distinctive IMFs and effectively extract fault information for bearing fault detection under different operating conditions. It has potential to be applied in bearing health monitoring in high-speed trains.

## Figures and Tables

**Figure 1 sensors-23-06392-f001:**
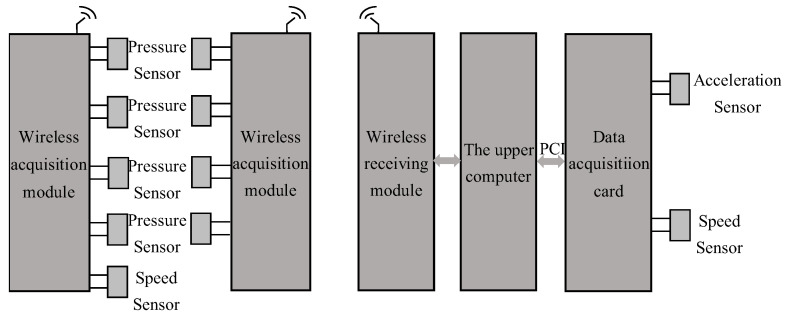
The flowchart of the information of the high-speed train DAQ system.

**Figure 2 sensors-23-06392-f002:**
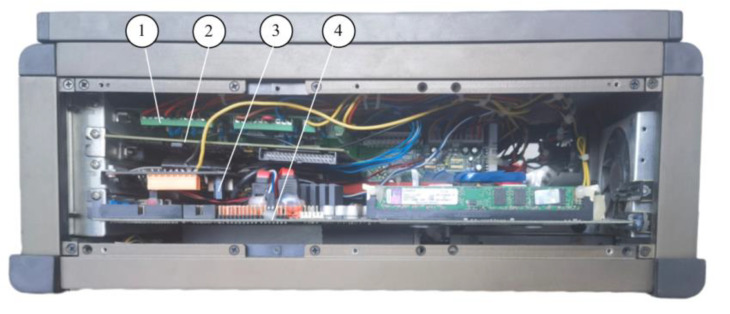
Mainframe assembly: (1) the signal conditioning board; (2) the data acquisition card; (3) wireless module; (4) CPU mastering.

**Figure 3 sensors-23-06392-f003:**
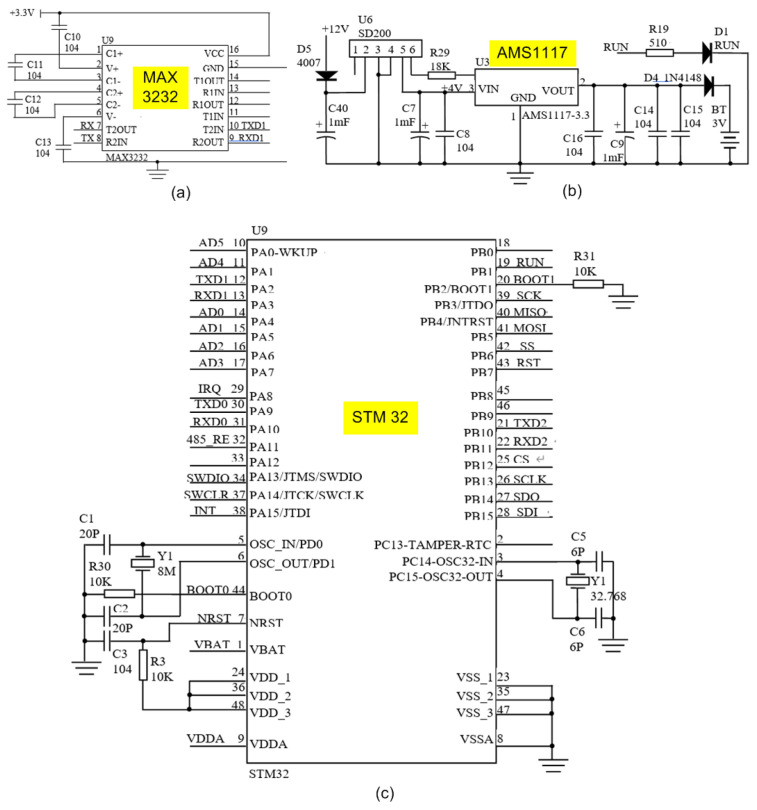
Embedded wireless acquisition module circuit schematic: (**a**) MAX3232 chip for RS232 communication; (**b**) AMS1117 voltage regulator (12 V to 3.3 V); (**c**) STM32 microcontrollers.

**Figure 4 sensors-23-06392-f004:**
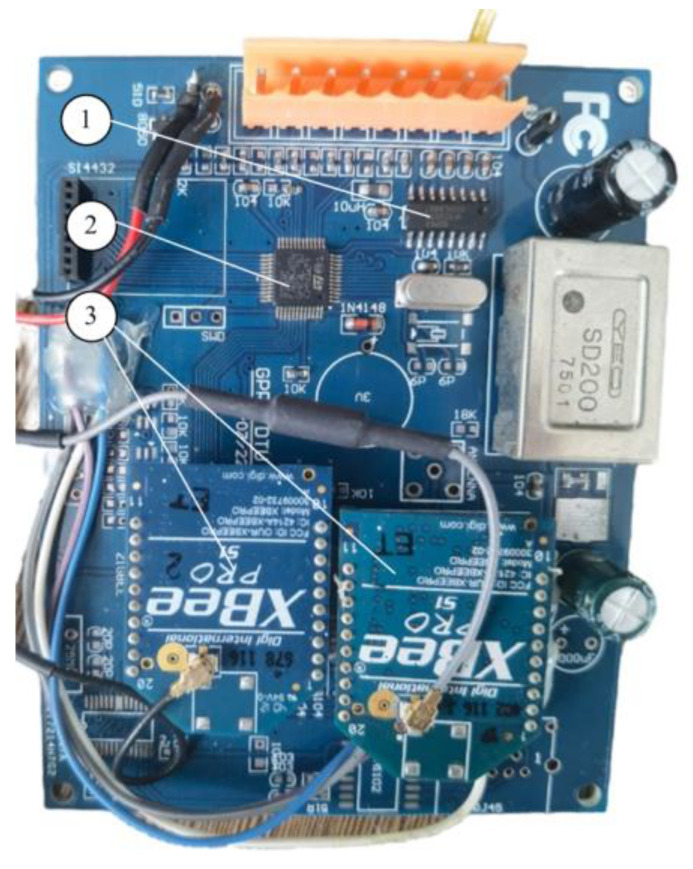
The developed embedded wireless module: (1) MAX3232; (2) STM32 microcontrollers; (3) wireless modules.

**Figure 5 sensors-23-06392-f005:**
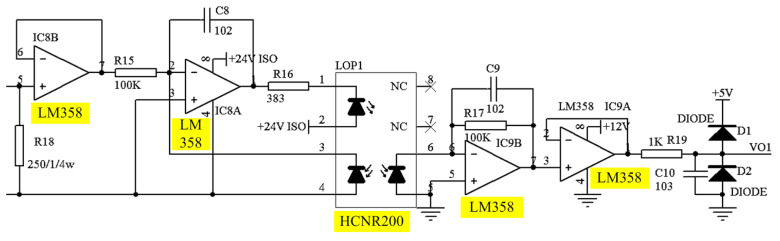
Signal conditioning board schematic: (1) LM358: Operational amplifiers; (2) HCNR200: linear analog photocoupler.

**Figure 6 sensors-23-06392-f006:**
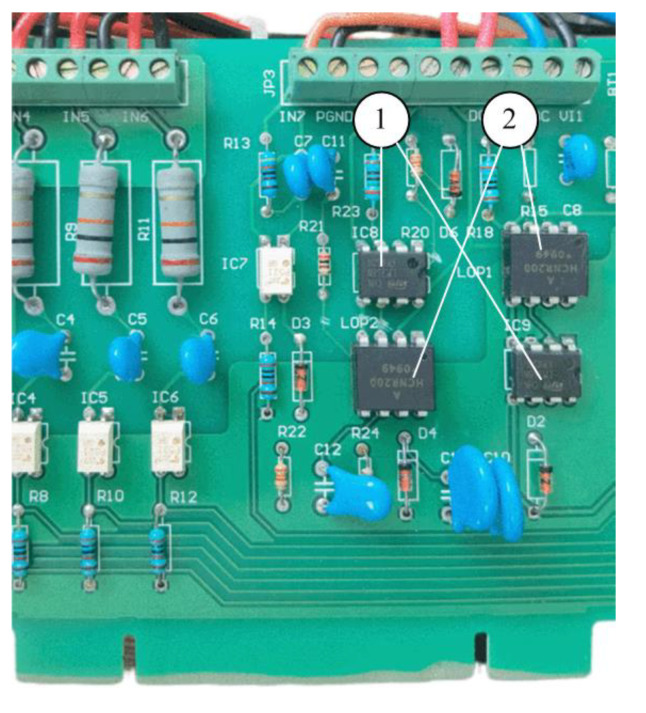
The developed signal conditioning board: (1) LM358; (2) HCNR200.

**Figure 7 sensors-23-06392-f007:**
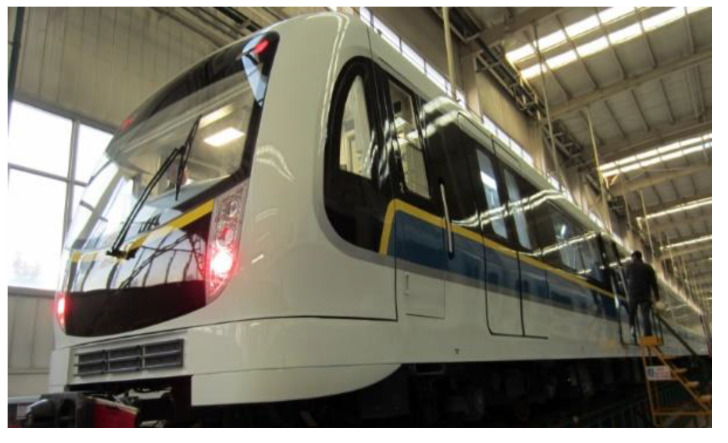
Experimental test in a high-speed train.

**Figure 8 sensors-23-06392-f008:**
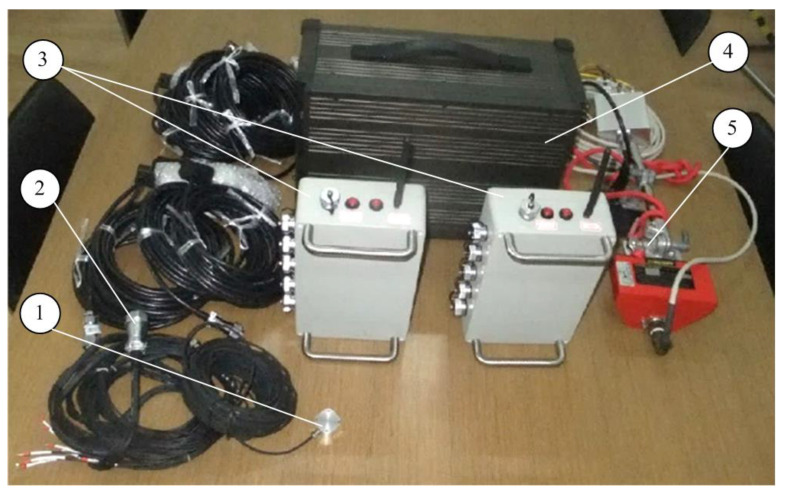
Part of the related DAQ hardware system: (1) pressure sensors; (2) accelerometers; (3) wireless collectors; (4) upper computer; (5) speed sensor.

**Figure 9 sensors-23-06392-f009:**
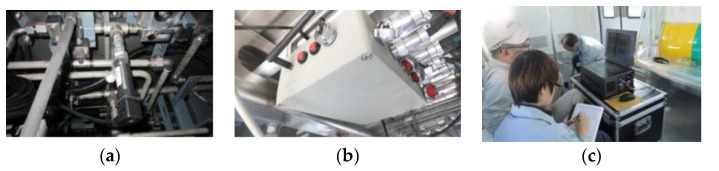
(**a**) The installed pressure sensor for measurement; (**b**) the installed wireless collector; (**c**) the commissioning of the DAQ processes.

**Figure 10 sensors-23-06392-f010:**
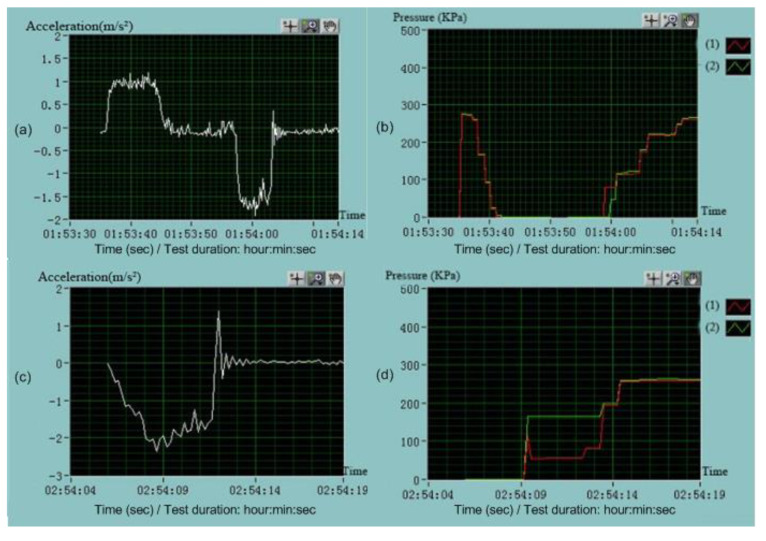
The developed DAQ and monitoring interface using Labview: (**a**,**c**) instantaneous acceleration curve; (**b**,**d**) part of the pressure signals from the second car; (1) the pressure from the first axle of the second car; (2) the pressure from the third axle pressure of the second car.

**Figure 11 sensors-23-06392-f011:**
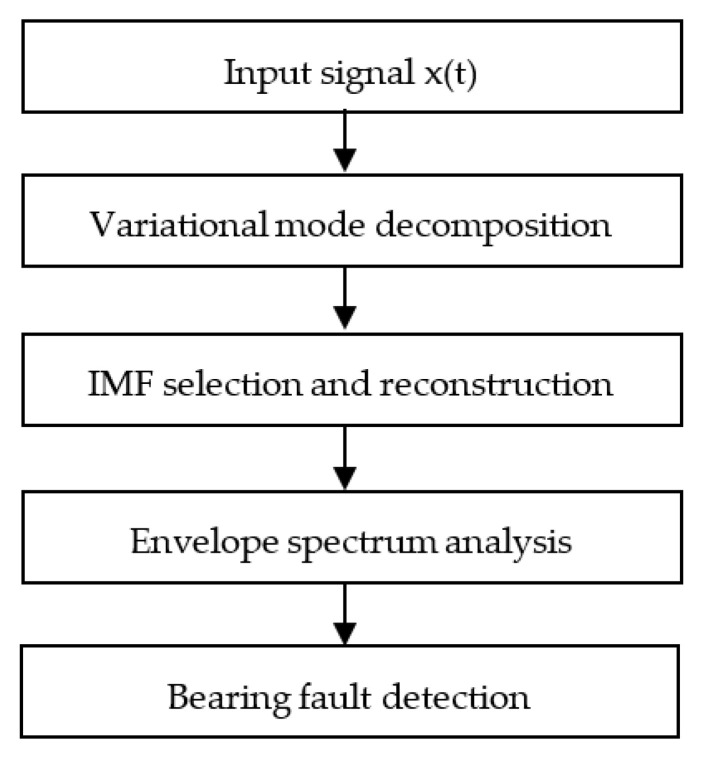
The flowchart of the proposed MCA technique.

**Figure 12 sensors-23-06392-f012:**
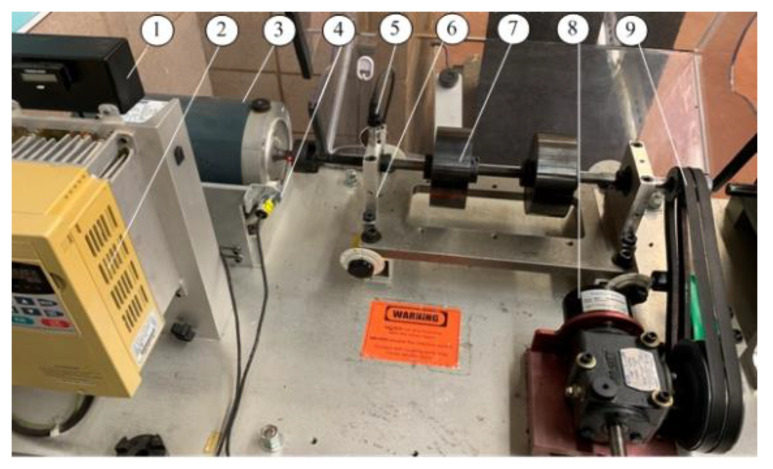
Experimental setup: (1) encoder display; (2) speed control; (3) drive motor; (4) optical encoder; (5) accelerometer node; (6) bearing housing; (7) misalignment adjustor; (8) variable load system; (9) belt drive.

**Figure 13 sensors-23-06392-f013:**
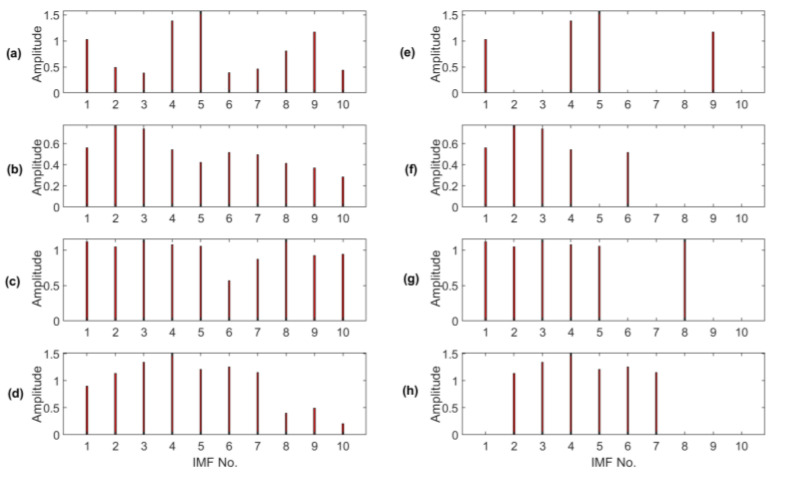
Comparison before and after IMF value selection corresponding to different bearing health conditions: (**a**,**e**) healthy bearing; (**b**,**f**) bearing with outer race defect; (**c**,**g**) bearing with inner race defect; (**d**,**h**) bearing with rolling element defect.

**Figure 14 sensors-23-06392-f014:**
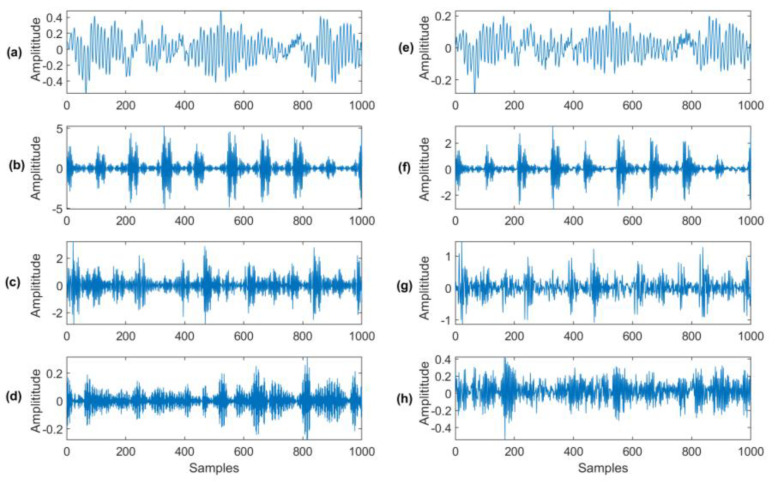
Parts of reconstructed signals and original signal with different health conditions: (**a**,**e**) healthy bearing; (**b**,**f**) bearing with outer race defect; (**c**,**g**) bearing with inner race defect; (**d**,**h**) bearing with rolling element defect.

**Figure 15 sensors-23-06392-f015:**
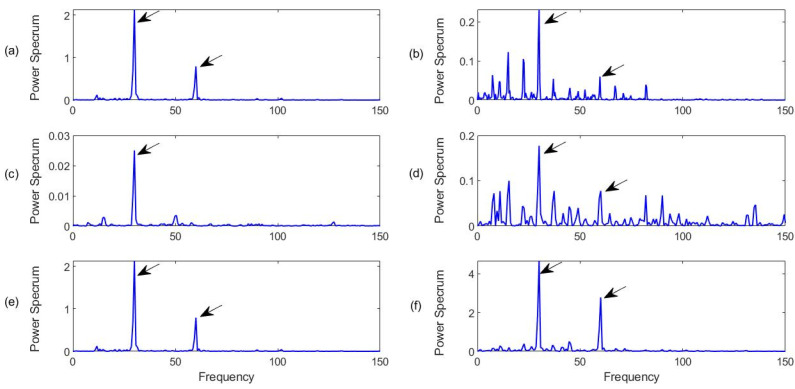
Comparison of processing results for a healthy bearing using the related techniques: (**a**) SII; (**b**) FBE; (**c**)HH-C; (**d**) HH-K; (**e**) NME; (**f**) MCA. Arrows specify characteristic frequency and its harmonics.

**Figure 16 sensors-23-06392-f016:**
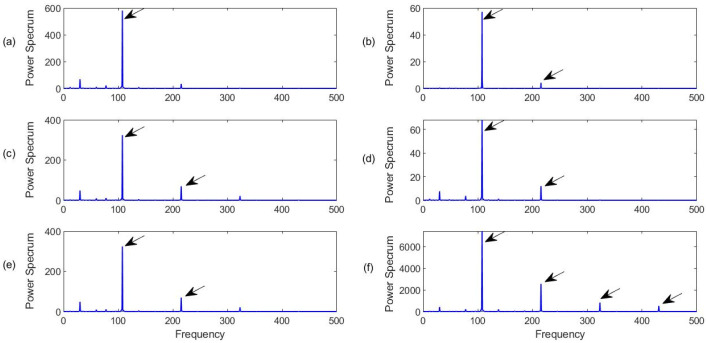
Comparison of processing results for a bearing with outer race fault detection using the techniques of (**a**) SII; (**b**) FBE; (**c**)HH-C; (**d**) HH-K; (**e**) NME; (**f**) MCA. Arrows specify characteristic frequency and its harmonics.

**Figure 17 sensors-23-06392-f017:**
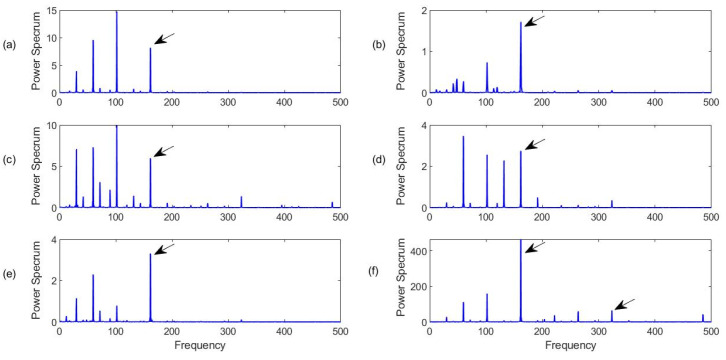
Comparison of processing results for a bearing with inner race fault detection using the techniques of (**a**) SII; (**b**) FBE; (**c**)HH-C; (**d**) HH-K; (**e**) NME; (**f**) MCA. Arrows specify characteristic frequency and its harmonics.

**Figure 18 sensors-23-06392-f018:**
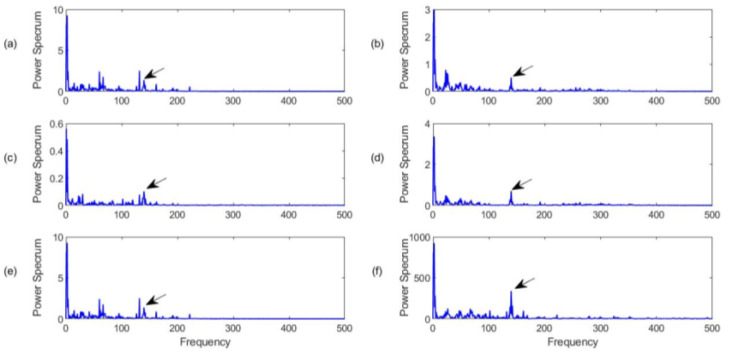
Comparison of processing results for a bearing with the ball fault detection using the techniques of (**a**) SII; (**b**) FBE; (**c**)HH-C; (**d**) HH-K; (**e**) NME; (**f**) MCA. Arrows specify characteristic frequency and its harmonics.

**Table 1 sensors-23-06392-t001:** Structural parameters of tested bearings.

Bearing Type	MB ER-10K
Inside diameter (mm)	25.001
Outside diameter (mm)	51.999
Thickness(mm)	15.001
Pitch diameter (mm)	39.034
Rolling element diameter (mm)	7.940
Number of rolling elements	9
Contact angle (degrees)	0

**Table 2 sensors-23-06392-t002:** The wireless collector reference specifications [17].

Bearing Condition	Characteristic Frequency (Hz)
Normal/healthy bearing	*f_r_*
Outer race defect	3.58 *f_r_*
Inner race defect	5.41 *f_r_*
Rolling element defect	4.71 *f_r_*

## Data Availability

The study did not report any data.

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
