# Peer review of "A New Monitoring Technology for Bearing Fault Detection in High-Speed Trains"

_sensors, 2023, doi:10.3390/s23146392_

Round 1
Reviewer 1 Report
This study proposed a new monitoring technology for bearing fault detection in high-speed trains. The paper is well-written and has a good structure. However, the article needs minor revisions, as summarized in the report below.
· The paper's novelties and innovative features should appear clearly in the abstract and at the end of the introduction.
· Figures 3 and 5 should be improved to be readable.
· Section 4.2 contains a good comparison (Figures 13 to 15), but a comparison with others (found in the literature) should be added.
· The conclusion should be revised.
The paper is well-written.
Author Response
Firstly the authors are very appreciative of this reviewer’s valuable comments and suggestions.
1. Reviewer: The paper's novelties and innovative features should appear clearly in the abstract and at the end of the introduction.
- Authors’ Response: Done as suggested. The paper's novelties and innovative features have been highlighted in the abstract and at the end of the introduction.
2. Reviewer: Figures 3 and 5 should be improved to be readable.
- Authors’ Response: Done as suggested. The main chips in Figures 3 and 5 have been highlighted in the revised manuscript.
3. Reviewer: Section 4.2 contains a good comparison (Figures 13 to 15), but a comparison with others (found in the literature) should be added.
- Authors’ Response: Done as suggested. Two more related techniques have been added to compare the performance of the proposed MCA technique: synchronous influence index method and frequency band entropy analysis method. The related comparison results are demonstrated in Figures 15 to 18 in the revised manuscript.
4. Reviewer: The conclusion should be revised.
- Authors’ Response: Done as suggested. The conclusion part has been revised.
Reviewer 2 Report
In this work, a new monitoring system is developed for bearing fault detection in high-speed trains. The authors have developed a system that has adequate application in practice.A new technique was developed for deciding on the state of the system, based on three signals collected through the measurement and acquisition system. The MCA technique enables a better spectral analysis of the signal compared to the existing ones. Perhaps the authors should have given it more importance in the analysis. I believe that the research presented in this article is very useful and applicable in practice.
Perhaps the authors should specify the software in which they implemented the measurement-acquisition system.
Author Response
Firstly the authors are very appreciative of this reviewer’s valuable comments and suggestions.
1. Reviewer: Perhaps the authors should specify the software in which they implemented the measurement-acquisition system.
- Authors’ Response: Done as suggested. We used Labview for measurement and data acquisition. It has been clarified in the last paragraph on page 7 in the revised manuscript. Figure 10 shows an example using Labview for data acquisition.
Reviewer 3 Report
This paper presents the development of a data acquisition system for vibration and an algorithm that uses variational modal decomposition and multiple correlation analysis of the acquired data to diagnose bearing faults in high-speed trains. The multiple correlation analysis is used to identify signal characteristic features, and the bearing fault is diagnosed by examining the frequency information on the envelope power spectrum. In my view, the paper is well presented, and technology has high applicability.
Some comments to improve the presentation are:
1. In the introduction can be mentioned that there are model-based methods for fault diagnosis, e.g., a Review of Convex Approaches for Control, Observation, and Safety of Linear Parameter Varying and Takagi-Sugeno Systems, Processes.
2. Please define all acronyms in first use, e.g., R&D is not defined. Also, check that all variables are defined.
3. Figure 10 shows plots from the application. If possible, include units for the Acceleration variable.
4. In Equation 7, what does the semicolon mean?
5. Please include units in all the plots.
6. A comparison of the fault detection algorithm with a recent and relevant approach is needed in the results section.
In general, the language is ok. Minor edits are needed.
Author Response
Firstly the authors are very appreciative of this reviewer’s valuable comments and suggestions.
1. Reviewer: In the introduction can be mentioned that there are model-based methods for fault diagnosis, e.g., a Review of Convex Approaches for Control, Observation, and Safety of Linear Parameter Varying and Takagi-Sugeno Systems, Processes.
- Authors’ Response: Done as suggested. Some model-based fault diagnosis methods have been mentioned in Introduction in paragraph 2 on page 2.
2. Reviewer: Please define all acronyms in first use, e.g., R&D is not defined. Also, check that all variables are defined.
- Authors’ Response: Done as suggested. R&D stands for research and development, which has been clarified in the revised manuscript.
3. Reviewer: Figure 10 shows plots from the application. If possible, include units for the Acceleration variable.
- Authors’ Response: Done as suggested. The units of the acceleration variables have been added in Figure 10.
4. Reviewer: In Equation 7, what does the semicolon mean?
- Authors’ Response: The semicolon stood for a variable separator. To prevent possible confusion, it has been changed to a comma in the revised manuscript.
5. Reviewer: Please include units in all the plots.
- Authors’ Response: Done as suggested.
6. Reviewer: A comparison of the fault detection algorithm with a recent and relevant approach is needed in the results section.
- Authors’ Response: Done as suggested. Two more related techniques have been added to compare the performance of the proposed MCA technique: synchronous influence index method and frequency band entropy analysis method. The related comparison results are demonstrated in Figures 15 to 18 in the revised manuscript.
Reviewer 4 Report
The article is devoted to the development of a monitoring system for bearing fault detection of high-speed trains based on vibration measurements and other data. For data analysis, multiple correlation analysis is used in combination with variational modal decomposition. The effectiveness of the proposed system is confirmed by experimental results.
On the whole, the article is well-formed and contains new and interesting results. There are minor comments on the article:
1. Line 244: The criterion of precision convergence is ε>0. However, the article does not present how ε is defined.
2. The abbreviation IMF first occurs on line 62, and the transcript is given only on line 68.
3. What is s.t. in formula 1?
4. The abbreviation NNE in the text is given without deciphering.
5. Line 391: It should be MCA instead of MAC.
These remarks are insignificant and do not concern the essence of the article. In general, the article is recommended for publication.
Proofreading is required. For example, line 359, 365: "it can see".
Author Response
The authors are very appreciated to this reviewer’s valuable comments and suggestions.
1. Reviewer: Line 244: The criterion of precision convergence is ε>0. However, the article does not present how ε is defined
- Authors’ Response: Corrected as suggested. ε is a small positive number. Its value is usually over (0, 0.01] (or 1%). ε = 0.001 is used in this work. The related comment has been added on page 10 in the revised manuscript.
2. Reviewer: The abbreviation IMF first occurs on line 62, and the transcript is given only on line 68.
- Authors’ Response: It has been corrected. The acronym of IMF has been defined in Abstract in the revised manuscript.
3. Reviewer: What is s.t. in formula 1?
- Authors’ Response: The s.t. in Equation (1) stands for “subject to”. To prevent possible confusion, it has been changed to “Constraints” in Equation (1) in the revised manuscript.
4. Reviewer: The abbreviation NNE in the text is given without deciphering.
- Authors’ Response: Corrected as suggested. NNE has been replaced by NME. The acronym NME has been defined in paragraph 6 on page 13, which NME is a combination of the NCM, MIA and ESC.
5. Reviewer: Line 391: It should be MCA instead of MAC.
- Authors’ Response: The typo MAC on page 15 has been corrected.
Round 2
Reviewer 3 Report
I am happy with the revised manuscript.
Minor edits.